# Overcoming Barriers in Glioblastoma—Advances in Drug Delivery Strategies

**DOI:** 10.3390/cells13120998

**Published:** 2024-06-07

**Authors:** Esther ter Linden, Erik R. Abels, Thomas S. van Solinge, Jacques Neefjes, Marike L. D. Broekman

**Affiliations:** 1Department of Cell and Chemical Biology and Oncode Institute, Leiden University Medical Center, 2300 RC Leiden, The Netherlands; e.ter_linden@lumc.nl (E.t.L.); e.r.abels@lumc.nl (E.R.A.);; 2Department of Neurosurgery, Leiden University Medical Center, 2300 RC Leiden, The Netherlands; t.s.van_solinge@lumc.nl; 3Department of Neurosurgery, Haaglanden Medical Center, 2512 VA The Hague, The Netherlands

**Keywords:** blood–brain barrier, glioblastoma, drug delivery, focused ultrasound, chemotherapy, nanoparticle

## Abstract

The world of cancer treatment is evolving rapidly and has improved the prospects of many cancer patients. Yet, there are still many cancers where treatment prospects have not (or hardly) improved. Glioblastoma is the most common malignant primary brain tumor, and even though it is sensitive to many chemotherapeutics when tested under laboratory conditions, its clinical prospects are still very poor. The blood–brain barrier (BBB) is considered at least partly responsible for the high failure rate of many promising treatment strategies. We describe the workings of the BBB during healthy conditions and within the glioblastoma environment. How the BBB acts as a barrier for therapeutic options is described as well as various approaches developed and tested for passing or opening the BBB, with the ultimate aim to allow access to brain tumors and improve patient perspectives.

## 1. Introduction

Cancer treatment has been improving fast, and new tumor treatment strategies like targeted therapies and immunotherapy have enlarged in the treatment field for many different cancer types. Unfortunately, not all cancer patients have benefited from these new strategies, and the treatment of primary brain tumors still highly relies on surgical resection, with dissatisfactory results [1,2].

Gliomas are diffuse-growing primary brain tumors that arise from glial progenitor cells. Glioblastoma (GBM) is the most common malignant primary brain tumor and accounts for 14.2% of all CNS tumors and 50.9% of all malignant brain tumors [3]. The standard of care for GBM consists of maximal safe neurosurgical resection, followed by postoperative radiotherapy with concomitant and adjuvant treatment with temozolomide (TMZ), a DNA alkylating cancer drug. Unfortunately, the effect of this treatment regimen is limited, and relapse is common, resulting in a median overall survival of only 15 months [4,5].

The only substantial change to this regimen has been the addition of Tumor Treating Fields (TTFs). Since 2014, this non-invasive technique of alternating electrical fields inducing anti-cancer effects via a multitude of biological processes is approved as an addition to the standard-of-care therapy [6]. The first clinical results imply a substantial improvement in overall survival of several months, which is now being tested more extensively [7]. The addition of TTFs and their improvement in overall survival are major steps in the treatment of GBM patients, but this simultaneously emphasizes the need for more effective treatments.

One of the key challenges in the treatment of glioblastoma is the blood–brain barrier (BBB), which limits the use of many therapeutic agents. TMZ is one of the few chemotherapeutics that is effective in GBM treatment. However, even TMZ delivery is somewhat limited by the BBB. The brain parenchymal concentration of TMZ is only 20% of the systemic dose, which is mainly attributed by efflux drug transporters at the BBB that prevent the brain access of many chemotherapeutic agents [8]. Also, other classes of anti-cancer treatments like antibody- and protein-based cancer therapy are unable to cross the BBB and are, therefore, ineffective in the treatment of GBM [9].

This is one of the main reasons why the many effective anti-cancer therapeutical agents have failed to develop as a standard of care for GBM [10]. Drug delivery strategies are, therefore, essential to increase treatment options for GBM patients.

Several strategies are currently being explored. In this review, we will first describe the current understanding of the BBB in the context of glioblastoma. Next, we will discuss recent advances in drug delivery: manipulation of the BBB, opening of the BBB, and local approaches circumventing the BBB. This will be completed through a discussion of the future prospects of GBM therapies.

## 2. Blood–Brain Barrier

The BBB comprises all vasculature in the brain and strictly regulates the exchange of oxygen and metabolites between the brain and the blood. Its protective function prevents neurotoxic agents and pathogens from entering the brain and generates an immune-privileged space that suppresses inflammatory responses that would be disastrous within this enclosed and highly specialized organ [11]. Concurrently, these protective features also prevent the adequate delivery of systemic chemo- and immunotherapies to the brain.

A system of cerebral endothelial cells (CECs) surrounded by pericytes, astrocytes, and perivascular microglia compose the neurovascular unit (NVU). Through coordinated intercellular signaling, the NVU regulates the functions of the BBB, like blood flow and waste clearance. Different cells within the NVU have specific functions. Structural support of the NVU is provided by the pericytes, while astrocytes are important in maintaining the integrity and functionality of the endothelial barrier [12], as well as facilitating communication with neurons [13] (Figure 1A). However, the main executors of the BBB are the CECs. To prevent the uncontrolled extravasation of blood products into the brain, CECs are firmly joined by a combination of intercellular tight junctions and adherens junctions, which conjointly inhibit all paracellular transport, with the exception of small or gaseous molecules [14]. Due to the restricted para-cellular transport, the passage of nutrients and other metabolites is regulated by trans-endothelial transport (Figure 1B).

Solute carriers (SLCs) are multimembrane spanning proteins responsible for the transport of solutes into and out of the endothelial cells by the mechanism of carrier-mediated transport (CMT). They are the largest group of membrane transporters, with nearly 500 family members in the human genome. They can transport substrates unidirectionally or bidirectionally according to the concentration gradient or against the gradient using active transport [15]. Examples of SLCs that facilitate the uptake of nutrients into the brain are glucose transporters (GLUT1/SLC2A1) and glutamate transporters (GLT-1/SLC1A2) [16]. Influx carriers have been targeted to facilitate the BBB crossing of small molecules or nanoparticles [17,18].

The main active carriers in the BBB are ATP-binding cassette (ABC) efflux transporters, which are primarily responsible for the efflux of many waste products like metabolites out of the brain (Figure 1B). Numerous drugs that would be able to passage endothelial cells through simple diffusion are substrates of these ABC transporters, resulting in a strongly limited penetration of these drugs into the brain. Doxorubicin is one such substrate of the ABCB1 efflux transporter and is pumped back into the blood with high efficiency, preventing its use in the treatment of brain tumors [19]. ABC-efflux transporters are the main executors for drug export from the BBB, and, as a consequence, inhibitors of ABC-efflux transporters have been developed to inhibit efflux of ABC-efflux transporter substrates [20].

Macromolecules such as lipoproteins and transferrin cannot be handled by solute carriers and are transported into the brain by transcytosis. Transcytosis is a multi-phase system that only occurs in polarized cells and consists of endocytosis, intracellular vesicle trafficking and exocytosis at the opposite site of a polarized cell. This can occur by membrane charge-mediated adsorptive transcytosis or by the binding of a ligand to its receptor, which mediates the endocytosis of the ligand and the ligand-associated compounds [21]. In non-polarized cells, the endosomes ultimately mature to lysosomes where substrates can be degraded. How ligands select transcytosis rather than follow the endosome-lysosome route is still being discussed [22]. Multiple receptors for transcytosis are highly expressed by the endothelial cells of the BBB and facilitate the delivery of important nutrients to the brain. Transferrin receptor (TfR), insulin and insulin-like growth factor receptors (INSR, IGF1), low-density lipoprotein receptor (LDL-R), and LDL-receptor-related protein (LRP) are some of the receptors mediating transcytosis at the BBB [23]. Receptor-mediated transcytosis is intensely studied as a basis for specific brain-targeted drug delivery systems of larger molecules and nanoparticles [24] (Figure 1B).

With immune-based therapies showing their therapeutic potential for many cancer types, the understanding of immune cell trafficking over the BBB is increasingly important. The brain is an immune-privileged area and, under physiological circumstances, only peripherally activated circulating T cells have the specific ability to cross the BBB [25]. How exactly the BBB provides this immune privilege is still under debate, but limiting leucocyte extravasation and tissue penetration seems to be an important part of its function. Leucocyte extravasation at the BBB takes place in the postcapillary venules and encompasses a multi-step process of initial tethering by E- and P-selectins, followed by rolling and slowing of the leukocytes. Upon sensing endothelial chemotactic cues with G-protein-coupled receptors, integrin-mediated arrest and, ultimately, leucocyte extravasation occur [26]. Contrary to endothelial cells in other organs, BBB endothelial cells lack constitutive P-selectin storage [27] and the continuous expression of atypical chemokine receptor 1 (ACKR1), resulting in a lower T-cell extravasation. After extravasation, a second barrier created by astrocytic endfeet prevents further penetration of T cells into the brain parenchyma [28,29]. These many barriers prevent the uncontrolled entry and activity of the peripheral immune system in the brain. How this affects cancer immunology therapies for brain tumors remains unclear but has to be elucidated to generate effective immunotherapy of brain tumors.

## 3. Blood–Brain Tumor Barrier and Drug Delivery

The BBB is a tightly regulated network. However, under pathological conditions, like trauma, neuroinflammatory diseases, or brain tumors, the integrity of the BBB can be disrupted [30,31,32]. In GBM, the growth of tumor cells and environmental changes within the tumor can modify the BBB. To distinguish normal BBB from abnormalities caused by GBM, the latter is often referred to as the blood–brain tumor barrier (BBTB). The development of the BBTB is a multifaceted process. First, the physical growth of tumor cells distorts the organization of the neurovascular unit. When the support and coordinated signaling of the pericytes and astrocytic feet are lost, many of the endothelial barriers including the tight junctions will be lost as well [33,34]. Second, the high tumor mass creates hypoxic areas that trigger angiocrine factors like vascular endothelial growth factor (VEGF) expression [35]. Not only do these angiocrine factors influence the pre-existing vasculature but they also induce neovascularization, consisting of a highly immature vasculature [36].

In combination, the disruption of the tight junctions and immature vasculature results in an leaky BBTB, allowing uncontrolled diffusion of molecules into the underlying brain tissue [33] (Figure 1C). This can be visualized by the presence of intratumoral-free fluid in contrast-enhanced MRI [37].

Together with other tumor characteristics such as an impaired lymphatic system [38,39], this leakiness of the BBTB results in a passive accumulation of blood-derived molecules within the tumor. This phenomenon of increased import and decreased outflow is labeled the enhanced permeability and retention (EPR) effect [40] and has been affiliated with increased local drug delivery in GBM [41,42].

However, there are a few reasons why we cannot just rely on the EPR effect for drug delivery. Firstly, and most importantly, the disruption of the BBTB is not a homogeneous process. GBM is a diffusely growing tumor that often displays a compact tumor core but has a broad infiltrating border, with tumor cells migrating between otherwise normal brain parenchyma. The disruption of the BBTB is most prominent in the tumor core but less obvious or even absent at the tumor periphery [43,44] (Figure 1A). Contrast-enhanced MRI supports these findings, with a lack of enhancement in the GBM periphery [45]. Secondly, ABC-efflux transporters are frequently upregulated in GBM, further inhibiting the brain penetration of many drugs in areas with an intact BB(T)B [46]. Thirdly, the disruption of the NVU and the loss of endothelial function have been linked to an altered perfusion of parts of the tumor core, actually resulting in a local limitation of drug delivery [13,47]. This suggests that tumors can locally affect the barrier function of the BBB, but this will be insufficient for optimal drug delivery. Therefore, new drug strategies need to consider overcoming the BBB to sufficiently treat the entire GBM, including their peripheral parts.

The effect of the BBTB on immune cell activation and extravasation for brain tissue entry is poorly studied. In principle, a loss of barrier function should allow for the tissue entry of (tumor-specific) immune cells or antibodies into the brain. However, some GBM characteristics have been reported to hamper an effective influx of leucocytes. For example, reductions in the expression of intracellular adhesion molecule 1 (ICAM1) and vascular cell adhesion molecule 1 (VCAM1) at the BBTB may prevent leucocyte tethering and extravasation [48]. Furthermore, cytokines expressed by the CECs can create an immunosuppressive environment that hampers leucocyte activation and extravasation [49,50]. In the case of checkpoint inhibitors that are used in the treatment of lung, melanoma, and colon cancer, it is unknown whether these antibodies need to be able to cross the BBB or whether peripheral lymphocyte activation is sufficient. However, durable responses of GBM or glioma to checkpoint antibody immunotherapy have not been observed in unselected patient groups. While some patient may have a more durable response, it is unclear whether this results from a more optimal BBTB, more immunogenic tumors, or a better tumor microenvironment [51].

## 4. Overcoming the BBB

The understanding of the BB(T)B is growing, and multiple methods have been developed to overcome these barriers for drug delivery. Here, we discuss therapeutic options to overcome these barriers through (1) manipulation of the BB(T)B, (2) chemical and physical opening of the BB(T)B, and (3) local delivery.

## 5. Manipulating the BBB

### 5.1. Efflux Transporter Inhibitors

The efflux of metabolites and neurotoxins is facilitated by a series of efflux transporters situated on the luminal and basal membrane of the BBB endothelium. Consequently, most small-molecule therapeutic agents are substrates of these efflux transporters [52]. The effect of efflux transporters on the concentration of drugs in the brain is demonstrated by knock-out studies in mice that show enhanced brain penetration of efflux transporter substrates when drug efflux transporters are absent [53,54]. The best known and studied efflux transporters are ATP-binding cassette super family B1 (ABCB1) and G2 (ABCG2) that export non-polar and less amphiphilic molecules, like lipids and steroid hormones, but also drugs like temozolomide, paclitaxel, and anthracyclines (ABCB1) or mitoxantrone and topotecan (ABCG2). The ABCC transporter subfamily members ABCC1, ABCC3, ABCC4, and ABCC5 export more polar molecules. However, there is substantial overlap within the efflux transporters for substrates, collaborating in restricting brain access and the effectivity of drugs [52].

Efflux transporters can be inhibited to prevent drug efflux to increase the brain accumulation of therapeutic agents. ABCB1 and ABGC2 inhibitors include cyclosporin A (first generation), valspodar (second generation), elacridar, and tariquidar (third-generation inhibitors). They have different mechanisms of inhibition that either prevent the activity of the ABCB1/C2 transporter or compete for substrate binding [55]. Of note, the drug export pumps are also expressed in other tissues such as the colon, where the inhibition of the pumps can support the uptake of orally provided drugs but can also cause unwanted side effects [56].

First- and second-generation efflux inhibitors failed clinically because of poor binding affinity and specificity to ABCB1 [57], resulting in a significant toxicity profile [58]. However, the third-generation ABCB1 inhibitor elacridar had a higher specificity and a lower toxicity and showed enhanced brain penetration of multiple substrate drugs, like gefitinib [54], vandetanib [59], paclitaxel [60], docetaxel [61], and others, in multiple rodent studies [62]. Phase I studies showed a safe toxicity profile, with minor side effects [63]. Unfortunately, severe toxicity was observed when elacridar was combined with chemotherapeutics like doxorubicin [64]. Moreover, elacridar failed to increase brain concentrations of erlotinib in patients [65].

It is important to note that the efflux pump inhibitors studied to date have not been developed to overcome the BB(T)B but to overcome multidrug resistance by the upregulation of the drug efflux pumps in other solid tumors. The clear differences between the preclinical models and human studies illustrate the need for PKPD studies and drug timing of activity to achieve concomitant arrival of the drug in question during efflux pump inhibition [66]. Another potential application of such inhibitors would be a combination of efflux inhibitors and chemotherapeutic agents delivered in BB(T)B-specific nanoparticles, which could prevent the efflux of drugs after tumor delivery by and release from the nanoparticles. In addition, this would increase BB(T)B specificity and should result in lower general toxicity. Manipulating the drug export mechanisms is, at present, not sufficient for efficiently getting drugs beyond the BBB, but their potential is not fully explored.

### 5.2. Nanoparticles

To improve systemic availability, therapeutic agents have been packaged into nanocarriers or nanoparticles (NPs). Enveloping and shielding of the therapeutic agents will improve the solubility, protect them from degradation, and prolong their circulation half-life time. There are many different types of nanoparticles, with many more different drug combinations. The most frequently used nanocarriers include liposome-based NP, micelles, dendrimers, polymeric NPs, and inorganic NPs (Figure 2A). Of these, liposome-based NPs are the most commonly used delivery agents and show good bioavailability, decreased degradation, and can encapsulate hydrophilic as well as hydrophobic drugs [67]. Nanoparticles are often decorated with polymer shrouds such as polyethylene glycol (PEG) that can prevent recognition by immune cells and, thus, prevent degradation [68]. Traditionally, NP-mediated delivery relied on increased systemic availably with passive tumor-specific accumulation due to the EPR effect.

Preclinical studies identified many promising candidate NPs with increased in vivo brain accumulation and improved tumor responses in GBM [69,70,71]. However, only a few NPs have been tested in patients. PEGylated liposomes encapsulating doxorubicin (Doxil-PEG-liposomes) have been studied since the 1990s and showed improved tumor concentrations in solid tumors and limited cardiotoxicity [72] (Figure 2B). These are now licensed for the treatment of Kaposi’s sarcoma, ovarian cancer, breast cancer, and multiple myeloma. However, a phase II clinical trial of Doxil PEG-liposomes, added to the treatment of GBM patients, did not show significant benefits in overall survival and progression-free survival [73]. Packaging may then decrease toxicity but not necessarily improve the effectiveness of chemotherapy in the case of brain tumors. Liposomal doxorubicin variants have been extensively tested in preclinical models with various alteration of the formulation of the liposome. Preclinical studies often found enhanced delivery of liposomal doxorubicin to the tumor, associated with an increase in survival. In a study with tumor-bearing rats, this translated to a 29% increase in median survival compared to non-treated animals [74]. When comparing treatment (three weekly doses) using free-doxorubicin versus liposomal doxorubicin, it was found that the life span was increased up to 189% when using liposomal doxorubicin compared to 126% when using free doxorubicin [75]. Biodistribution studies further revealed that 48 h post-injection, liposomal doxorubicin was present at a 5-fold higher level (at a concentration of 10 ug doxorubicin per gram tumor) in the tumor compared to free doxorubicin [75]. These results were encouraging to pursue this in the clinic. However, several phase 2 clinical trials found that while the use of liposomal doxorubicin was well tolerated, it did not increase the progression-free survival as well as the overall survival [73,76]. Unfortunately, like many other clinical trials, these studies did not report or measure the absolute doxorubicin concentration in the tumor. Because of the lack of data, no conclusion can be drawn on the actual cause of treatment failure within these clinical trials.

The progressive insights into the heterogeneity of the BBTB and the EPR may explain the disappointing effects of classical nanoparticles on brain tumors. One option to increase BBB passage is to coat NPs with ligands of BBB-located receptors or carriers to facilitate transcytosis into the brain (Figure 2A). In this context, apolipoprotein receptors represent one of the receptor classes that are regularly targeted. Low-density lipoprotein receptors (LDL-R) and LDL-R-related proteins (LRP) are highly expressed on the brain endothelium and regulate the uptake of lipids and cholesterol-containing particles [23]. Angiopep-2 is an oligopeptide derived from the Kunitz domain of aprotinin and exhibits high LPR1 binding efficiency and improved transcytosis. This oligopeptide has been coupled to a series of drugs to support their entry across the BBB. The drug–ligand conjugate ANG1005, containing three paclitaxel residues linked to angiopep-2, showed benefits in a phase II clinical study of patients with recurrent brain metastasis from breast cancer [77]. Angiopep-2 has now been coupled to organic and inorganic NPs to, for example, increase the delivery of chemotherapeutic drugs like doxorubicin [78,79]

Another LDL-R and LPR1 receptor ligand is apolipoprotein E (ApoE). ApoE is a lipoprotein that facilitates the transport of cholesterol and other lipids as part of VLDL and LDL particles. NPs decorated with PEG or polysorbate 80 will passively absorb ApoE in the bloodstream, facilitating LDL-R- and LPR1-mediated transcytosis [80]. However, direct coating of NP with ApoE and especially small ApoE peptides can potentiate transfer across the BBB [80]. Combined treatments of ApoE peptide-coated polymers capsulating the cytolytic enzyme granzyme B (ApoE-PS-GrB) and immune-stimulating CpG oligonucleotides (ApoE-PS-CpG) were used to induce immunogenic cell death (ICD) of the targeted cells and stimulate the maturation of dendritic cells within the tumor (Figure 2B). These immunoadjuvants are typically administered via intracranial injection or convection enhanced delivery (CED) due to a lack of effectiveness when administered systemically. This study illustrates that combining these polymer devices with information required to pass the BBB would allow for systemic treatment [81]. It also shows the potential of constructing more advanced nanoparticles for drug delivery into the protected environment of the brain.

Transferrin is a serum iron carrier protein that binds to the transferring receptor 1 (TfR1), regulating the transport of iron ions across the BBB [82]. The high serum levels of transferrin create competition with transferrin-coated NPs for the receptor. TfR1 binding peptides (T7 and T12) [83], monoclonal antibodies (OX26 and RVS10), and antibody fragments have been developed that target TfR1 at a non-ligand binding site, thereby avoiding competition with serum transferrin. Preclinical studies improved brain drug delivery of peptide- or antibody-decorated NPs, resulting in prolonged survival of mice with experimental GBM [84,85,86,87] (Figure 2B).

Even solute carrier proteins have been targeted for the BBB crossing of NPs. While most NPs are too large for carrier-mediated transport, binding of NPs to carriers can trigger their internalization and, ultimately, their transcytosis across the BBB [17]. Glucose transporter 1 (GLUT1) is highly expressed on the BBB endothelium. As a comparison, GLUT1 expression is approximately 100-fold higher than the transferrin receptor [88]. A 2-deoxy-D-glucose-modified NP system can use GLUT1 to increase the intracranial tumor accumulation of paclitaxel [88]. Glutathione (GSH) transporters are another target of carrier-mediated transport. GSH-conjugated PEGylated liposomes enveloping doxorubicin (GSH-Doxil) showed the delivery of this drug across the BBB in preclinical studies [88]. How these ligand conjugates are selected for transcytosis rather than degraded in lysosomes is not completely elucidated, and this part of the process can still be optimized [24].

Nanoparticles are becoming more sophisticated, often showing a combination of tumor-specific as well as BBB-targeted ligands to sufficiently increase brain tumor accumulation while decreasing side effects. Their versatility is not only in their surface decoration but also in the different therapeutic agents carried by NPs. Optimization of these combinations will be essential to arrive at improved treatment options for GBM.

Unfortunately, several issues have to be considered in NP-based treatments. While drug delivery to defined tissues can be significantly improved in comparison to free drugs, nanoparticles still suffer from clearance and poor pharmacokinetics. Peptide-decorated nanoparticles may be unstable as the peptides are swiftly degraded [89] but, simultaneously, proteins, peptides, or other molecules can promote serum protein absorption, resulting in a protein corona. This protein corona can significantly alter NP properties, their systemic availability, and, most importantly, their ability to bind their target [90,91]. Indeed, nanoparticle behavior is highly affected by protein corona formation in vivo [92]. Moreover, optimizing ligand density and affinity is still complicated. Increased ligand density improves receptor recognition but also increases off-target effects or decreases transcytosis due to receptor saturation [82]. At the same time, improved ligand affinity promotes NP internalization but can also reduce the release of the content from intracellular compartments. pH sensing ligand cleavage has been proposed to improve endosomal compartment escape [93].

Beyond the composition of the NP, the mode of administration also affects the tissue distribution of NPs. Overall, the administration of NPs can be achieved by intravenous injection, intra-tumor injection, and intranasal administration [94]. These different modes of administration have various advantages or disadvantages. First, intravenous administration is standard for systemic administration through the blood circulation. However, the NPs still have to pass the BBB, and the entry and delivery of cargo at other tissues obviously induce side effects. Second, intra-tumoral injection is an invasive mode of administration. The obvious advantage is the direct local delivery of the NP, resulting in a better drug distribution in the tumor. In addition, this reduces the side effects of drugs outside the tissue area of interest. This application is most often used for inorganic NPs that cannot be given systemically [95]. Lastly, intranasal administration is an easy-to-use system for patients. Intranasal delivery reduces systemic side effects and may bypass the BBB through diffusion through the olfactory mucosa and connective tissue suitable for local delivery. However local side effects and inefficient penetration of NPs into the brain and tumor are also serious limitations [96].

Altogether, modified NPs have improved the potential of delivering drugs into the brain, but the further optimization of various features will be essential to arrive at safe and active GBM treatments.

## 6. Chemical and Physical Opening of the BBB

### 6.1. Tight Junction Disrupters

The strongest barriers within the BBB are intercellular tight junctions of the CEC that almost completely prevent the paracellular extravasation of blood-derived molecules. The high-molecular-weight kininogen proteolytic peptide bradykinin can regulate these tight junctions. The bradykinin analog RMP-7 was used to open the BBB as early as 1986 [97]. Though preclinical and phase II studies showed increased passage of molecules across the BBB [98,99], a randomized, double-blind placebo-controlled phase II study comparing carboplatin alone or in combination with RMP-7 showed no clinical benefit for RMP-7 [100]. Interestingly, preclinical research showed that bradykinin production in GBM promotes macrophage activity and modulates GBM progression [101]. As a result, bradykinin receptor antagonists are now being proposed as new therapeutic options for GBM.

Tight junctions of CSCs can also be disrupted by the release of small-molecule adenosine. Regadenoson, a subtype A2A adenosine receptor activator, mediated a 60% increase in TMZ brain concentrations in non-tumor-bearing rats [102]. Unfortunately, no detectable change in BBB permeability was observed in patients treated with Regadenoson [103]. Strikingly, analogous to bradykinin, the biology is always more complicated. Adenosine receptors also have a role in tumor progression, and adenosine receptor antagonists are now also proposed as targets for anti-cancer therapy [104].

### 6.2. Hyperosmolar Agents

Mannitol is a hyperosmolar agent that, when administered, will result in dehydration and subsequent shrinkage of the endothelial cells, thus temporarily widening the gap between cells and promoting the transcellular diffusion of molecules into the brain [105]. Because of the short time window of BBB opening (between 10 and 40 min in humans [105]), mannitol is often combined with intra-arterial chemotherapeutics. Both the non-selective opening of the BBB as well as the invasive technique necessary for delivery of drugs into a GBM limit the popularity as a mode of BBB opening in the context of GBM, and clinical advances have not been reported [105]. However, mannitol in BBB-disrupting therapies may have benefit when used in combination with less neurotoxic chemotherapies [106] or viruses [107].

### 6.3. Focused Ultrasound

Another route of possible passage of the BBB by cancer drugs is to mechanically open the BBB by focused ultrasound (FUS). The two main types of FUS use either high- or low-intensity FUS. The principle of FUS rests on focusing ultrasound on a small, defined area. High-intensity focused ultrasound will result in a local increase in temperature up to 55 °C and is used for thermal ablation and contributes to rapid tissue necrosis [108]. This type of FUS is unsuitable for opening the BBB. To achieve temporal opening of the BBB, a lower-intensity ultrasound is used. Applying low-intensity FUS (LIFU) in combination with intravenous administration of microbubbles, indeed, increased vascular permeability. This increase in permeability is caused by the expansion and oscillation of the microbubbles, which mechanically loosen the tight junctions between the endothelial cells [109]. In addition, the sonication will also result in the release of specific cytokines, contributing to reduced BBB integrity and increased passage of molecules from the blood circulation into the brain [110].

Preclinical and clinical trials have been conducted to study the enhanced delivery of drugs in glioblastoma tissue by FUS. FUS indeed increases the brain delivery of chemotherapeutics, such as temozolomide and doxorubicin, as well as substrates as sizeable as antibodies [111] (Figure 3).

To demonstrate the potency of FUS to increase the delivery of chemotherapeutic agents into a glioblastoma, several groups have studied the anti-cancer drug doxorubicin—which normally does not cross the BBB—in combination with FUS [112,113]. Liposomal doxorubicin in combination with FUS did not result in severe neurotoxicity, as compared to convection-enhanced delivery [114]. Using a syngeneic mouse model where FUS was combined with the administration of doxorubicin showed increased concentrations of drug in the GBM tumor and improved survival [115]. Similar effects were obtained with FUS in combination with liposomal doxorubicin [116,117,118,119]. Improved extravasation of doxorubicin by FUS was confirmed by intravital multiphoton imaging of doxorubicin [120]. Interestingly, combining thermosensitive liposomal doxorubicin with FUS demonstrated that high-intensity FUS could also increase the delivery of doxorubicin to the site of the tumor [121]. Local physical disruption of the BBB creates options to finally enter brain tumors with cancer drugs that otherwise would not exclude from these tumors.

In the clinic, several trials have been initiated to study the effectiveness of FUS for drug delivery in glioblastoma. Different FUS devices are being tested, including, SonoCloud, MRgFUS, and NaviFUS. The first device is an implantable device that can be activated through a transdermal needle connection attached to an external control unit. The second device uses high-intensity ultrasound, while NaviFUS can be used in combination with a CT scanner to navigate FUS to the location for manipulation. A direct comparison as to the most efficient method of FUS-induced drug delivery has not yet been reported, but the current studies illustrate progress in this area for better GBM treatment options.

## 7. Can We Ignore the BBB?

The direct injection of therapeutic agents within the tumor or the tumor cavity after surgical resection is an obvious solution to circumvent the problem of drug passage across the BBB. However, a single injection of drugs directly into the brain tumor tissue was not sufficient to induce lasting anti-tumor responses. The drug distribution in the tumor tissue may be limited and, furthermore, hampered by high intratumoral pressure. Multiple direct drug delivery techniques have been developed to overcome these limitations.

### 7.1. Convection-Enhanced Delivery

Convection-enhanced delivery (CED) is a combination of one or multiple local injections in combination with a positive pressure gradient, resulting in an improved tissue distribution compared to injection alone [122,123]. Considering the highly infiltrative growth pattern of glioblastoma, drug distribution within the brain parenchyma is important to effectively treat the more distant parts of the tumor. At the same time, an improved drug distribution could result in increased toxicity, and tumor specificity becomes a more important factor to take into consideration. New trials studying CED in combination with tumor-specific therapeutic agents like nanoparticles or oncolytic viruses are, therefore, of special interest [124]. Desjardins et al. reported the safety of intratumoral injection of the oncolytic polio/rhinovirus recombinant (RVSRIPO) that showed no neurovirulent potential but is specifically targeted to tumor cells and dendritic cells [125]. These first results allowed for progression to a phase II study. Another application of CED is the local delivery of cellular therapy. Chimeric antigen receptor T cells (CAR-T) have been used to induce the anti-tumor immune response in different tumors outside the brain. Even though T cells are able to cross the BBB, no clinical results in GBM have been reported [126]. Atik et al., therefore, proposed the application of local delivery of CAR-T cells in GBM [127] (Figure 3).

### 7.2. Cavity Treatments

Brain tumors may also directly be approached by using the brain cavity that is left after brain tumor surgery as a drug depot. Biodegradable wafers that cover the resection cavity and deliver local (high-dose) chemotherapy have been developed to enable long-term local treatment. Gliadel wafers with the DNA methylating anti-cancer drug carmustine are approved chemotherapeutic implants for the treatment of GBM. Even though Gliadel wafers showed prolonged overall survival compared to radiotherapy alone [128,129], the effectiveness compared to the current standard of care (radiotherapy combined with systemic TMZ) is unclear [130]. Gliadel wafers are, therefore, hardly used in clinical practice. Wafers with different therapeutic agents are currently being examined. Cerebra biodegradable wafers with the active compound (Z)-n-butylidenephthalide ((Z)-BP), a small molecule that is aimed to overcome TMZ resistance, were reported safe in a phase I trial and showed therapeutic effects in a phase II trial [131]. Many other (polymer-based) cavity placement techniques are being developed that all focus on prolonged local delivery of drugs following their slow release for the wafer. The dosage and timing of drug release from these cavity techniques are not always clear, as well as the distribution of drugs into the surrounding (tumor) tissue [132]. The prolonged drug release without the need for additional surgical intervention does provide a major advantage, and designing the optimal biomaterial for the rationally selected therapeutic agent may further increase the application of these wafers in GBM treatment [133] (Figure 3).

## 8. Conclusions: Where Do We Stand?

The blood–brain barrier (BBB) is a major treatment-limiting hurdle for the effective delivery of drugs to brain tumors. Various approaches have been developed to allow for efficient access of drugs to these brain tumors. These include the manipulation of drug transporters on the endothelial cells of the BBB with the use of NPs to increase the uptake and crossing of drugs across the BBB. Chemical or mechanical disruption of the BBB can facilitate the uptake of previously unusable drugs or simply bypassing the BBB by direct injection or by wafers placed in the post-surgical resection cavity.

The study of brain drug delivery is clearly becoming more relevant in the treatment of GBM patients. It is for this reason that there is an extensive number of (preclinical) studies that focus on a multitude of different approaches, as reviewed here. Even though some of the drug delivery approaches have existed for years, the development within these techniques still leads to valuable applications within this field. Nanoparticles are becoming extensively more sophisticated not only in their targeting but also their therapeutic content. In combination with a better understanding of transcytosis, this may support the first step, essential in brain entry: the passage through the endothelial layer into the brain. Other developments include the neurosurgical advancements, facilitating safer and more versatile options for local delivery and non-invasive techniques like FUS, allowing for diffusion across the BBB of previously unusable drugs. Even though development is still ongoing, we believe that the combination of these specific drug delivery techniques together with advancements in the knowledge and therapeutic options of GBM could lead to a breakthrough in GBM treatment.

## 9. Discussion: How to Approach Brain Tumors for Optimal Therapy

At present, none of the drug delivery techniques are used as standard of care. When evaluating all announced clinical trials, only a fraction focus on drug delivery strategies, with the main focus on local delivery through CED. One of the reasons may be the high failure rate of earlier clinical trials, where the various strategies were applied. While preclinical research provided a multitude of promising treatment strategies, in-patient trials failed to reproduce these findings. The limitations of the studies and, therefore, this review lie in the high heterogeneity of experimental models for GBM and the experimental setups used within this field. This also makes it challenging to perform a comparative analysis on the different strategies. Furthermore, the quantitative measurement of drug penetration and concentration within different areas of the tumor is challenging, especially in human patients. This leaves a gap in our understanding of the efficiency of the different drug delivery systems, which is required to further improve treatment options for brain tumors.

An alternative strategy would be the search for cancer drugs that cross the BBB. Repurposing well-established drugs that have been developed for the treatment of other diseases may be an effective method of drug discovery, with the benefit of streamlining the passage of these agents for their new treatment indications. Particularly, drugs that are known to cross the BBB are interesting for glioblastoma treatment. Psychotropic drugs are interesting potential drug candidates because of their known brain penetration, in addition to the extensive experience with these. Interestingly, several preclinical studies have identified anti-tumor effects of antipsychotics for various cancer types, including brain tumors [134,135,136]. One of the major drawbacks of repurposed drugs is their unknown interactions with other drugs and the possibility of unexpected adverse effects. Furthermore, the mechanism of action for anti-tumor activity may be different than their primary function, as reported by Weissenrieder et al. for a Dopamine D2-like receptor antagonist that elicits cytotoxicity in a calcium-dependent, non-D2 receptor-dependent manner [137]. Consequently, significantly higher concentrations of the compound are needed to achieve cytotoxicity, and this may introduce toxic side effects. Achieving relevant concentrations of drugs to eliminate tumors under mildly toxic conditions will be clinically challenging. Many other drugs are of interest for repurposing in the treatment of GBM (see review of Ntafoulis et al., [138]). Unfortunately, taking these repurposed often off-patent drugs into clinical trials has proven financially and logistically difficult, as the patient population is small, and the trials and drugs are poorly supported. Yet, upcoming in silico approaches including AI can help to prescreen interesting candidate drugs to select the GBM drug candidates with the highest likelihood of being successful in clinical trials. If successful, this can substantially improve the preselection of interesting repurposed drug candidates [139].

Targeted therapies are an excellent approach for the treatment for many cancers to increase tumor toxicity and limit negative side effects. One option is the use of antibody–drug conjugates. Antibody–drug conjugates are a combination of a monoclonal antibody with a chemical linker to connect a cytotoxic drug to the antibody. With this approach, cytotoxic drugs can be specifically targeted to unique antigens expressed on tumor cells. Many GBM tumor cells express a mutated form of EGFR, EGFRvIII [140]. This highly tumor-specific antigen has been exploited to target GBM cells. So far, different cytotoxic drug (such as mafodotin, DM1, and PBD) combinations with monoclonal antibodies for EGFRvIII have been tested in GBM. Unfortunately, no clinical effects have been observed using these antibody conjugates [141], likely due to their inability to cross the BBB due to their size of up to 160 kDa [142]. However, using these antibody–drug conjugates in combination with a BB(T)B opening technique like FUS would be the next step to render these compounds usable for GBM patients.

While the blood–brain barrier is a unique morphological and functional feature to the brain only, a similar barrier is found in the testis. The blood–testis barrier serves a similar function to create an immune-privileged environment and protect tissue beyond the barrier from harmful agents in the circulation. The composition of the blood–testis barrier is different from the blood–brain barrier but shares similarities, like the presence of tight junctions and efflux transporters. Furthermore, isolated testicular relapse occurs at similar frequencies to tumors in the central nervous system (CNS) [143]. This suggests a comparable drug barrier-inhibiting effective treatment of local testicular cancers. This similarity could provide the opportunity to use drugs and technologies effective for one tissue-specific tumor type for the other tumor type.

Crossing the BBB is only the first yet critical step in effective drug delivery to GBM and other brain tumors, and knowledge on the distribution and pharmacodynamics of the drug at the site of the tumor will be crucial to determine and increase its anti-cancer potential. In-depth studies on the behavior of therapeutic agents within the tumor and surrounding brain tissue will be the next step to get a grip on drug efficacy in glioblastoma and other brain tumors and select the best approach for GBM patients, who are eagerly awaiting better therapies.

## Figures and Tables

**Figure 1 cells-13-00998-f001:**
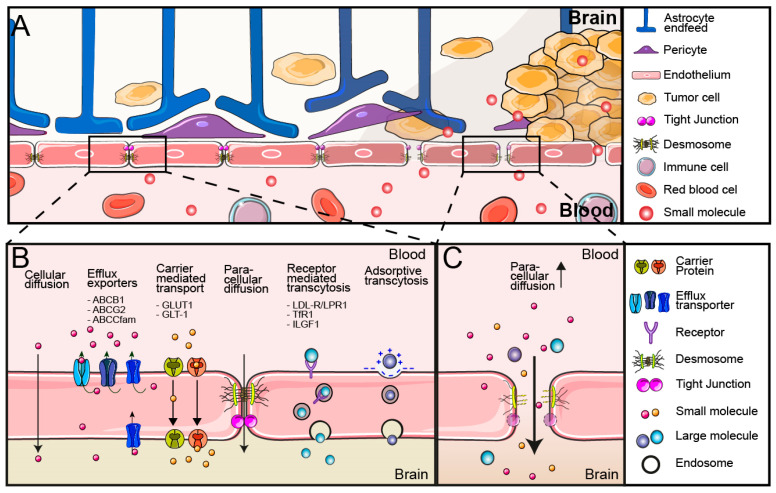
Blood–brain barrier and blood tumor brain barrier. Schematic overview of the blood–brain barrier and blood–tumor barrier. (**A**) Under physiological conditions, the neurovascular unit, consisting of the endothelial cells, pericytes and astrocytes in communication with neurons, orchestrates an intact BBB. During tumor formation, the neurovascular unit is disrupted, resulting in a dysfunctional and leaky BBTB. BBTB disruption and leakiness are heterogenous processes displayed by a decreased disruption of the tight junctions and areas of intact BBB at the peripheral border of the tumor. (**B**) The blood–brain barrier under physiological conditions has intact endothelial cells that are connected via tight junctions and adherence junctions, preventing most paracellular transport. Mechanisms of cellular transport are displayed, including (1) cellular diffusion for some small lipophilic molecules, (2) efflux-transporters like ABCB1, ABCG2 and other ABCC family members that transport substrates out of the endothelium and back in to the circulation, (3) carrier-mediated transport for the uptake of small molecules and ions including glucose facilitated by carrier proteins GLUT1/SLC2A1 and glutamate by GLT-1/SLC1A2, (4) paracellular diffusion which is largely prevented by the tight junctions and adherence junctions, (5) receptor-mediated transcytosis that facilitates uptake of macromolecules such as lipoproteins via LDL-R/LPR1 receptor, transferrin by TfR1 and insulin by INSR, and (6) membrane charge-mediated adsorptive transcytosis. (**C**) The blood–brain tumor barrier with increased paracellular transport due to a disorganization of the neurovascular unit and loss of tight junctions. ABCB1/G2/C, ATP-binding cassette B1/G2/C; BBB, blood–brain barrier; BBTB, blood brain tumor barrier; GLUT, glucose transporter 1, GLT-1, glutamate transporter 1; IGF1, insulin-like growth factor 1; INSR, insulin receptor; LDL-R, low-density lipoprotein receptor; LPR1, low-density lipoprotein receptor-related protein 1; LSC1A2, solute carrier family 1 member 2; LSC2A1, solute carrier family 2 member 1; TfR1, transferrin receptor 1. Figure contains modified images from Servier Medical Art (https://smart.servier.com/ accessed on 1 May 2024) under a Creative Commons Attribution 4.0 Unported License.

**Figure 2 cells-13-00998-f002:**
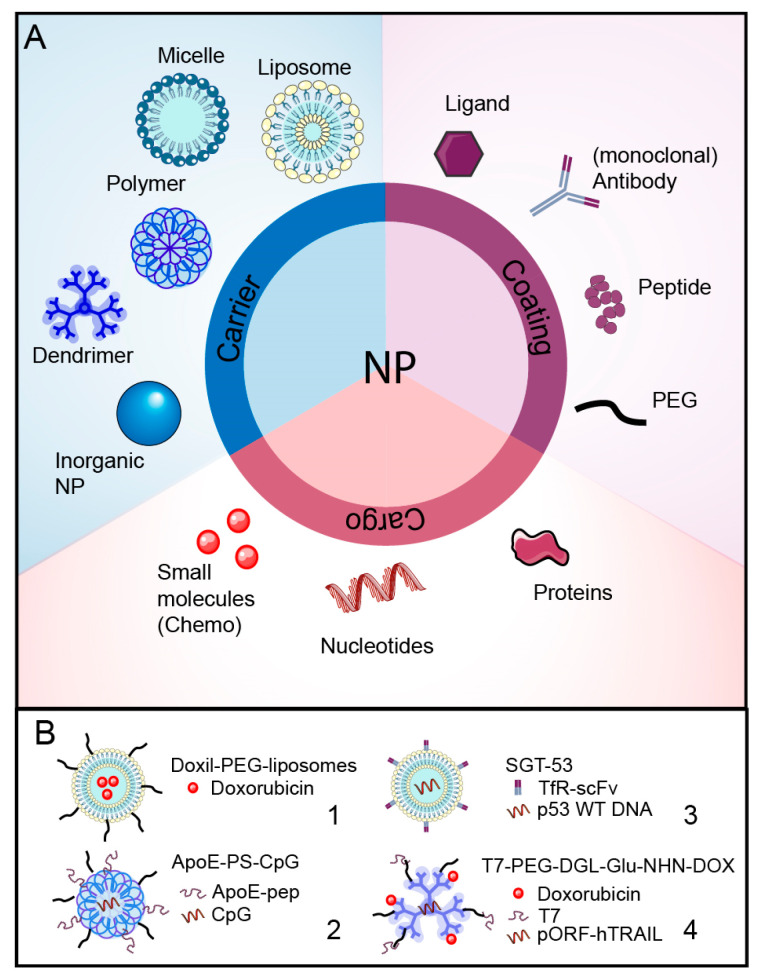
Nanoparticle formulations. (**A**) Overview of the variety in NP formulations based on the carrier, the therapeutic cargo and its coating. Coating can be used to avoid immunosurveillance (PEG) or increase cell-type specificity, promote transport or act as therapeutic agents (ligand, antibodies and peptides). Most often, chemotherapeutics is the cargo of choice, but nucleotides are emerging as cancer-specific therapeutic agents. (**B**) Visual examples of four NP formulations used in glioblastoma studies and discussed in this review. Doxil-PEG-liposomes are based on a liposome carrier, PEG-coating and doxorubicin as therapeutic cargo (1); ApoE-PS-CpG consists of polymers coated with PEG and an apolipoprotein peptide to target the LPR-1 receptor for BBB crossing. The therapeutic cargo is CpG-dinucleotide, which induces immune cell activation (2). SGT-53 is a liposome-based NP enveloping a DNA plasmid encoding wildtype p53. Anti-transferrin receptor single-chain antibody fragments (TfR-scFV) facilitate binding to the transferrin receptor and BBB transcytosis (3). T7-PEG-DGL-Glu-NHN-DOX is the most sophisticated NP displayed here. This dendrimer is coated by transferrin receptor binding peptide T7. Furthermore, it carries doxorubicin together with a gene drug, tumor necrosis factor-related apoptosis-inducing ligand-encoding plasmid (pORF-hTRAIL) that induces apoptosis by binding to death receptors 4 and 5 on tumor cells. To increase tumor specificity, doxorubicin is linked via an acid-sensitive linkage to release doxorubicin only in an acid tumor environment (4). Figure contains modified images from Servier Medical Art (https://smart.servier.com/ accessed on 1 May 2024) under a Creative Commons Attribution 4.0 Unported License.

**Figure 3 cells-13-00998-f003:**
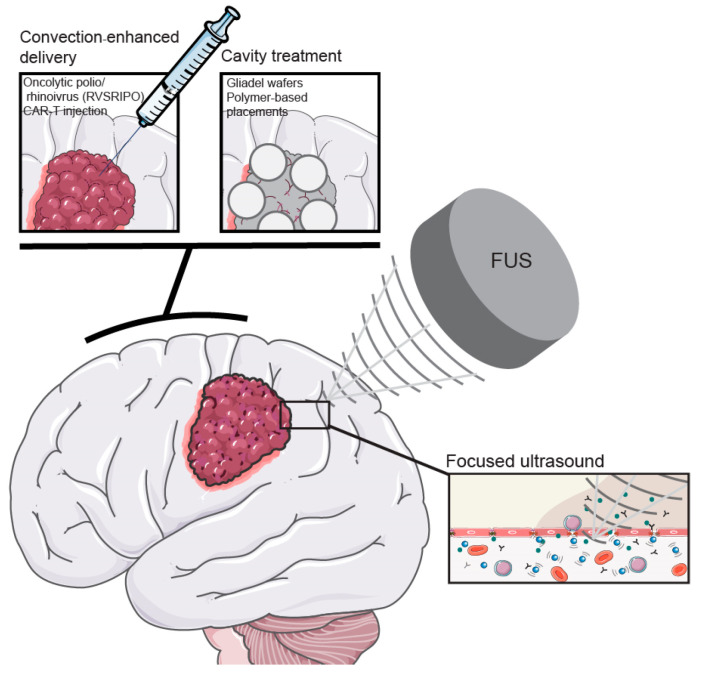
Clinical interventions aimed at the mechanical delivery of drugs across the BBB. Surgical and non-invasive methods to deliver drug over the BBB include the use of (1) FUS, (2) convection-enhanced delivery and (3) cavity treatment. (1) With the non-invasive methods of FUS, ultrasound opens the BBB temporarily, resulting in an increased concentration of drugs in the brain tumor. (2) With convection-enhanced delivery, therapeutic agents—including CAR-T cells, MSCs and oncolytic viruses—are directly injected into the tumor. (3) Following resection of the tumor, wafers carrying drugs are placed in the cavity to treat the residual tumor. Figure contains modified images from Servier Medical Art (https://smart.servier.com/ accessed on 1 May 2024) under a Creative Commons Attribution 4.0 Unported License.

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
