# Peer review of "Overcoming Barriers in Glioblastoma—Advances in Drug Delivery Strategies"

_cells, 2024, doi:10.3390/cells13120998_

Round 1

Reviewer 1 Report

Comments and Suggestions for Authors

This review article focuses on the effect of the BBB on therapeutic options are reported as well as various approaches developed and tested for passing the BBB with the ultimate aim to improve patient perspectives. To overcome the barriers for drug delivery and exploit the specific alterations in the BB(T)B. They also discuss therapeutic options to overcome these barriers by; 1) manipulation of the BB(T)B, 2) opening of the BB(T)B; and 3) local delivery. Overall, it appears updated and comprehensive. Below are several points need to be addressed:

1.     How to distinct BBB and BBTB still a challenge. So far, we assume that exploit the specific alterations could in the BB(T)B, however, no molecular biomarkers or really clear images were shoed in Figure 1, you propose.

2.     Since only TMZ drug is available for cross BBB, new uses for old drugs Identified from anti-psychotic drug as an anti-glioblastoma agent, is it easy to find out druggable for GBM? You may discuss this strategy.

3.     On Figure 3. Clinical interventions to deliver drug behind the BBB. Tumor-treating fields (TTF) has been approved on 2011 by FDA Tumor-treating fields (TTFields), a noninvasive anticancer therapeutic modality, has been rising as a fourth treatment option for GBMs You may also discuss this method.

Ref: Rominiyi, O., Vanderlinden, A., Clenton, S.J. et al. Tumour treating fields therapy for glioblastoma: current advances and future directions. Br J Cancer 124, 697–709 (2021). https://doi.org/10.1038/s41416-020-01136-5

4.     Lines 266-268, you describe “Combined treatment of ApoE peptide-coated polymers capsulating the immune stimulating agents granzyme B (ApoE-PS-GrB) and CpG oligonucleotide (ApoE-PS-CpG) were used to generate immunogenic cell death (ICD) and simultaneously activate dendritic cells”. Could dendritic cells across the BBB? I speculate that ICD and microenvironment could attribute to brain tumors due to the context of BBB,

5.     Finally, treating relapsed brain tumors, and the BBB adds an additional layer of complexity

Reviewer 2 Report

Comments and Suggestions for Authors

Broekman and colleagues present a literature review on "Overcoming Barriers in Glioblastoma: Advances in Drug Delivery Strategies." It is a timely and valuable compilation of literature, and the authors have done a great job. I recommend publishing this manuscript as it is. However, one section about targeting glioblastoma with antibody-drug conjugates and its perspective will enhance the value of the manuscript. If the authors can add this, it would be great.

Reviewer 3 Report

Comments and Suggestions for Authors

Glioblastoma (GBM) stands as the most prevalent and lethal malignant primary brain tumor. The current standard treatment protocol for GBM involves maximal safe neurosurgical resection, subsequent postoperative radiotherapy, and concurrent/adjuvant temozolomide (TMZ) therapy—an alkylating cancer drug targeting DNA. Despite these efforts, the impact of this treatment approach remains limited, resulting in a median overall survival of 15 months. A significant hurdle in GBM treatment is the blood-brain barrier (BBB), which restricts the efficacy of numerous therapeutic agents. While TMZ can penetrate the BBB, other drugs have not shown substantial survival benefits beyond the standard regimen. Enhancing drug delivery strategies becomes paramount to expand treatment avenues for GBM patients. This review summarizes therapeutic options and emerging technologies designed to breach the blood-brain barrier for effective chemotherapy delivery.

I have several comments to improve the manuscript:

In Figure 1, I couldn't locate 1A and 1B. Are the authors referring to "Up" and "Down" instead? It would be helpful for the authors to briefly mention the roles of ABCB1, Glut1, LDL/LPR1, etc., in the Figure legend to facilitate easier understanding of the Figure.

On Page 4, line 142, the authors mentioned that GBM growth creates a leaky blood-brain tumor barrier (BBTB), which can be demonstrated with contrast-enhanced MRI. Do the authors know why chemotherapy still cannot penetrate the BBTB? There is also a typo in "contrast-enhanced MRI" that needs correction.

In Figure 2, could the authors provide examples of drug names for each group (nanoparticles, coating, cargo) to enhance clarity?

In the conclusion, could the authors mention which drug transporters are currently used in patient treatment as part of the standard of care? Are these drugs only in clinical trials, or are they already established in clinical practice?

Reviewer 4 Report

Comments and Suggestions for Authors

The review manuscript by Abels et. asl., with the title of the work “Overcoming Barriers in Glioblastoma; Advances in Drug Delivery Strategies”.

Some points that need to be considered about the manuscript

- In the introduction, being able to include updated statistical data on the incidence and survival rates of glioblastoma would help to highlight the urgency and relevance of the research discussed, providing a more robust context for the need for new therapeutic approaches. Furthermore, further discussion on the limitations of current treatment strategies could clarify where exactly new approaches are needed, establishing a clear basis for the innovations proposed in the manuscript, emphasizing the importance of the BBB in glioblastoma tumors.

- It would be important to present experimental evidence, such as data from clinical and preclinical studies that demonstrate the challenges and effective strategies to overcome BBTB. The current discussion could be expanded to include comparisons with tumor barriers in other cancer types, providing a comparative view that could suggest cross-sectional approaches applicable to glioblastoma.

- One aspect that could be addressed is the relationship between administration routes and nanoparticles for the treatment of glioblastoma.

- About the manuscript discussion could be improved by integrating the findings with existing literature, where it needs to be highlighted how they expand current understanding. It would also be important to include a more detailed analysis of remaining challenges and potential solutions, along with some recommendations for future research directions, making the discussion more robust and targeted.

- tables could be included with studies related to the BBB and nanoparticles and routes of administration with comparative data.

- Write about the limitations of the study.

Comments on the Quality of English Language

about English is adequate

Round 2

Reviewer 4 Report

Comments and Suggestions for Authors

All suggested revisions have been addressed, and I therefore recommend that the manuscript be published.

Comments on the Quality of English Language

Minor editing of English language required